


# Oxidative potential in rural, suburban and city centre atmospheric environments in Central Europe

Máté VÖRÖSMARTY[1], Gaëlle UZU[2], Jean-Luc JAFFREZO[2], Pamela DOMINUTTI[2],
Zsófia KERTÉSZ[3], Enikő PAPP[3], and Imre SALMA[4]

[1] Hevesy György Ph. D. School of Chemistry, Eötvös Loránd University, Budapest, Hungary
[2] University of Grenoble Alps, IRD, CNRS, INRAE, Grenoble, France
[3] Laboratory for Heritage Science, Institute for Nuclear Research, Debrecen, Hungary
[4] Institute of Chemistry, Eötvös Loránd University, Budapest, Hungary

*Correspondence to*: Imre Salma (salma.imre@ttk.elte.hu)

**Abstract.** Oxidative potential (OP) is an emerging health-related metric which integrates several physicochemical properties of particulate matter (PM) that are involved in the pathogenesis of the diseases resulting from the exposure to PM. Daily $PM_{2.5}$-fraction aerosol samples collected in the rural background of the Carpathian Basin and in the suburban area and centre of its largest city of Budapest in each season over one year were utilised to study the OP at the related locations for the first time. The samples were analysed for particulate matter mass, main carbonaceous species, levoglucosan and 20 chemical elements. The resulted data sets were subjected to positive matrix factorisation to derive the main aerosol sources. Biomass burning (BB), suspended dust, road traffic, oil combustion, vehicle metal wear and mixed industrial source were identified. The OP of the sample extracts in simulated lung fluid was determined by ascorbic acid (AA) and dithiothreitol (DTT) assays. The comparison of the OP data sets revealed some differences in the sensitivities of the assays. In the heating period, both the OP and PM mass levels were higher than in spring and summer, but there was a clear misalignment between them. In addition, the heating period-to-non-heating period OP ratios in the urban locations were larger than for the rural background by factors of 2–4. The OP data sets were attributed to the main aerosol sources using multiple linear regression with the weighted least squares approach. The OP was unambiguously dominated by BB at all sampling locations in winter and autumn. The joint effects of motor vehicles involving the road traffic and vehicle metal wear played the most important role in summer and spring, with considerable contributions from oil combustion and resuspended dust. In winter, there is temporal coincidence between the most severe daily PM health limit exceedances in the whole Carpathian Basin and the chemical PM composition causing larger OP. Similarly, in spring and summer, there is a spatial coincidence in Budapest between the urban hotpots of OP-active aerosol constituents from traffic and the high population density in central quarters. These features offer possibilities for more efficient season-specific air quality regulations focusing on selected aerosol sources rather than on PM mass in general.



**1 Introduction and objectives**

Poor air quality caused by high concentrations of particulate matter (PM) is one of the most severe public health concerns for humans worldwide (e.g. Lelieveld et al., 2015, 2020; Bondy, 2016; Cohen et al., 2017). Its acute and chronic effects, such as lung, cardiovascular and cerebrovascular diseases have been documented in both epidemiological and toxicological studies (e.g. Donaldson et al., 2005; Valavanidis et al., 2008; Apte et al., 2015; Riediker et al., 2019; Kelly and Fussell, 2020).

Due to the chemical, physical and biological complexity of ambient aerosol particles, their dynamic character and possible synergisms among air pollutants, a sophisticated interplay of multiple factors is involved in the pathogenesis of the diseases resulting from the exposure to PM. The main factors can involve: 1) mass concentrations of $PM_{2.5}$ or $PM_{10}$ size fractions, 2) amounts of potentially toxic chemical components such as transition and heavy metals, polycyclic aromatic hydrocarbons (PAHs), soot and specific organics, 3) certain chemical speciation forms such as Cr(VI) versus Cr(III), 4) lung bioaccessibility of critical constituents, 5) surface reactivity of particles, 6) number concentrations of very small particles such as ultrafine particles or engineered nanomaterials, 7) shape and morphology of particles such as for asbestos or silica and 8) active components of biogenic origin such as bacteria, viruses, pollens and moulds or with radioactivity such as radon progeny. Therefore, it cannot be expected that a single or a few aerosol metrics broadly express the induced biological responses. In the first approximation, PM mass is often selected from these factors as a simplistic metric, and it can be refined by further particle properties.

One of the most important biological mechanisms by which PM induces adverse health effects is causing an oxidant-antioxidant imbalance in the respiratory system at the cellular level (Kelly and Mudway, 2003; Borm et al., 2007; Kelly and Fussel, 2012, 2015; Cassee et al., 2013; Valavanidis et al., 2013). This is called as oxidative stress. It is related to 1) stimulating cells to uncontrolled production of excess reactive oxygen species (ROS) endogenously, e.g. directly by Fenton-type reactions of redox-active aerosol components in the human body or indirectly through biotransformation, e.g. of PAHs or 2) inefficient elimination of ROS by the antioxidant defence system of the body. These can lead to inflammatory processes that increase the risk for various diseases and can result in biological aging and apoptosis (Ayres et al., 2008; Verma et al., 2012; Gao et al., 2020). The capacity of PM to invoke oxidative stress is quantified by its oxidative potential (OP). This integrates several factors of the particle properties 1−8 listed above. Numerous epidemiological studies suggest that the OP can be one of the central quantities that is responsible for specific acute health effects such as emergency treatment of asthma and congestive



heart failure and that largely explains the underlying biological bases of toxicity (e.g. Bates et al., 2015;
Kelly and Fussel, 2015; Abrams et al., 2017; Yue et al., 2018; Daellenbach et al., 2020; Dhalla et al.,
2000; Zhang et al., 2022; Baumann et al., 2023).

As a result, there has been a substantial and increasing scientific interest in the measurement
improvements of OP using (biological) in vivo, in vitro cellular and in vitro acellular assays, and in the
identification of the aerosol components and sources closely related to OP (e.g. Cho et al., 2005; Künzli
et al., 2006; Boogaard et al., 2012; Verma et al., 2014, 2015; Kelly and Fussel, 2015; Fang et al., 2016;
Calas et al., 2017; Weber et al., 2018; Shahpoury et al., 2021; Borlaza et al., 2021b, 2022; Lionetto et al.,
2021; Zhang et al., 2022). The OP is frequently measured by acellular assays for exogenous ROS, in
which the PM extract or the particles directly cause a consumption rate of some antioxidants such as
ascorbic acid (AA) or of some chemical surrogates to cellular reducing agents, e.g. dithiothreitol (DTT).
The quantifications are generally based on spectrophotometry. More sophisticated detection methods
which directly determine the ROS production are also available (Katerji et al., 2019).

The most frequently used assays were compared for $PM_{10}$-fraction aerosol samples considering the
chemical composition of particles as well (Calas et al., 2018; Lionetto et al., 2021). It was concluded that
the assays strongly correlated with each other but were not equivalent. All assays were somewhat specific,
and no consensus has been reached on the "best assay" nor on a standardised methodology for each assay
(Weber et al., 2021; Zhang et al., 2022).

The main common features of the assays are that 1) they exhibit different responses to various groups of
ROS-generating compounds and their bioavailability, 2) their sensitivity depends on the partner reaction
compound to ROS, and 3) they show nonlinear response to PM mass concentration (Charrier et al., 2016;
Fang et al., 2016; Calas et al., 2017; Shahpoury et al., 2021). A large number of PM constituents were
identified to influence the OP. The DTT assay responds sensitively to ROS produced by organic
compounds and indirectly by soluble transition metals, mainly Cu(II), Mn(II) and Fe(II), and can be also
influenced by synergetic or antagonistic effects between some chemical components (Charrier and
Anastasio, 2012; Bates et al., 2019; Shahpoury et al., 2021; Borlaza et al., 2022). The AA assay was
shown to express large sensitivity to transition metals and some specific organics such as quinones
(Künzli et al., 2006; Godri et al., 2011; Visentin et al., 2016).

It is important to extend the studies on this emerging health-related metric to cities and regions in the
world. The knowledge on the OP for a large part of Central Europe, namely the Carpathian Basin, is



deficient or missing (Szigeti et al., 2015). The major objective of this study was to present, discuss and
interpreted the OP data determined by AA and DTT assays in PM$_{2.5}$-fraction aerosol samples collected in
parallel in central Budapest, its suburban area and rural background within the Carpathian Basin in each
season over one year. We also investigated the spatiotemporal dependencies, and identified the main
aerosol sources of OP. The study can contribute to our general knowledge on the OP as well.

## 2 Methods

### 2.1 Sample collections

The samplings in the rural background were performed at the K-puszta station (N 46° 57' 56", E 19° 32'
42", 125 m above sea level, a.s.l.), which represents the main plain part of the basin (Salma et al., 2020a).
Budapest, with ca. 2.2 million inhabitants in the metropolitan area, is the largest city in the region. Its
suburban environment was characterised by collections at the Marczell György Main Observatory (N 47°
25' 46", E 19°10' 54", 138 m a.s.l.) of the Hungarian Meteorological Service (Salma et al., 2022). This is
in a southeast residential part of the city. The samplings in the city centre were accomplished at the
Budapest platform for Aerosol Research and Training (BpART) Laboratory (N 47° 28' 30", E 19° 03' 45",
115 m a.s.l.), which represents an average atmosphere of the city centre (Salma et al., 2016).

Three identical high-volume sampling devices equipped with PM$_{2.5}$ inlets (DHA-80, Digitel, Switzerland)
were deployed at the sites (Salma et al., 2020a). The collection substrates were prebaked quartz fibre
filters with a diameter of 150 mm (QR-100, Advantec, Japan). Daily aerosol samples were taken starting
at midnight of local time. The samples corresponding to air volumes of 720 m$^3$ were collected in parallel
with each other over semi-consecutive days in October 2017 (autumn), January 2018 (winter), April 2018
(spring) and July 2018 (summer). The total numbers of the filters were 56 at the rural site, 59 in the
suburban area, and 28 in the city centre. The samples evenly spread among the four seasons. In addition,
one field blank was taken at each location and in each season. The filters were wrapped in preheated Al
foils, sealed in plastic bags and stored at a temperature of <−4 °C until the analyses. The samples represent
a gradual transition from the central part of a large continental European city through its suburban area to
its regional background in all seasons.

### 2.2 Analyses and data treatment

Particulate matter mass was determined by gravimetry (Cubis MSA225S-000-DA, Sartorius, Germany,
sensitivity of 10 μg). The blank-corrected PM mass data were above the limit of quantitation (LOQ),
which was 1 μg m$^{-3}$.




Filter punches were analysed by thermal-optical transmission method using a laboratory OC/EC analyser
(Sunset Laboratory, USA) adopting the EUSAAR-2 thermal protocol (Cavalli et al., 2010). All blank-
corrected organic carbon (OC) and elemental carbon (EC) data were above the LOQ, which were 0.38
and 0.04 µg m$^{-3}$, respectively. Filter pieces were analysed for levoglucosan (LVG) by gas
chromatography−mass spectrometry (Varian 4000, USA) after trimethylsilylation (Blumberger et al.,
2019). All blank-corrected LVG data were above the LOQ, which was 1.2 ng m$^{-3}$.

Parts of the filters were analysed by particle-induced X-ray emission spectrometry for S, Cl, K, Ca, Ti,
V, Cr, Mn, Fe, Co, Ni, Cu, Zn, As, Br, Rb, Sr, Zr, Ba and Pb using an external beam of protons with an
energy of 2.35 MeV and a current of 20 nA (Aljboor et al., 2022). The obtained spectra were evaluated
by the GUPIXWIN program. The filters were treated as thin layer samples. For S, Cl, K, Ca, the self-
absorption effects of quartz filters were corrected for (Chiari et al., 2018).

Concentrations of EC and OC from fossil fuel combustion and from biomass burning (BB), namely $EC_{FF}$
and $OC_{FF}$, $EC_{BB}$ and $OC_{BB}$, respectively and of OC from biogenic sources ($OC_{BIO}$) were previously
estimated by a coupled radiocarbon-LVG marker method (Salma et al., 2020a). Secondary organic carbon
(SOC) was also assessed previously using the EC tracer method for primary OC (Salma et al., 2022).
These results were used as supplementary data in interpretations.
**2.3 Determination of oxidative potential**
Specified filter areas were extracted in a simulated human respiratory tract lining fluid solution composed
of Gamble's solution with dipalmitoylphosphatidylcholine (DPPC; the major phospholipid of lung
surfactant; Calas et al., 2017, 2018). The extractions were carried out by vortex agitation at 37 °C for 1
h. The overall procedure represents conditions which are close to the respiratory system. Isoconcentration
extracts with 10 µg ml$^{-1}$ of PM mass were obtained for all samples to overcome possible nonlinear OP
response of PM concentrations (Charrier and Anastasio, 2012; Calas et al., 2017).

The OP of the extracts was measured without filtration by two single-compound in vitro acellular assays,
i.e. AA and DTT assays. These two methods are widely used for OP determination (e.g. Calas et al., 2018;
Daellenbach et al., 2020; Lionetto et al., 2021; Shahpoury et al., 2021). However, there is a fundamental
difference between them regarding their underlying chemical mechanisms (Charrier and Anastasio,
2012). The quantifications were based on plate-reader spectrophotometry (Tecan Infinite M200 Pro,
Switzerland) in MilliQ water for AA, and in a phosphate buffer with a physiological pH value of 7.4 after



adding 5,5'-dithiobis(2-nitrobenzoic acid), with readings taken at different specified reaction times (Calas
et al., 2018; Borlaza et al., 2021b, 2022). The possible transition metal contamination of the buffer was
removed by Chelex 100 resin to reduce the background oxidation. The consumption rates of the AA or
DTT were obtained from the simple linear regression of the absorbance values in time at 265 and 412 nm,
respectively. The coefficients of determination $R^2$ for the regression were >0.90 when <70 % of the initial
amount of the reagent was oxidised. Matrix absorbance was considered, and the quality assurance of the
determinations was performed by positive control tests (Borlaza et al., 2021b). The limits of detection
(LODs) for the AA and DTT assays were set at three times the standard deviation (SD) for the blank
extracts and were typically 0.008 and 0.0014 nmol min$^{-1}$, respectively. The experimental protocols were
described in more detail previously (Calas et al., 2017, 2018).

The OP data measured by the AA and DTT assays were normalised to PM$_{2.5}$ mass ($m$) or sampled air
volume ($V$) and are denoted as OP$_{AA,m}$, OP$_{DTT,m}$, OP$_{AA,V}$ and OP$_{DTT,V}$. The consumption rates normalized
to $V$ are often considered to have a closer relationship to human exposure, while those normalized by $m$
are regarded as a measure of the inherent toxicity of PM (Weber et al., 2018; Yu et al., 2019).
**2.4 Mathematical models**
Source apportionment modelling was accomplished to identify and quantify the main aerosol sources
using positive matrix factorisation (PMF, US Environmental Protection Agency, version 5.0; EPA, 2017).
It decomposes the sample data matrix into a linear combination of two matrices: a daily factor contribution
varying in time and factor profiles by minimizing the critical compound parameter Q (Paatero and Tapper,
1994; Hopke, 2016, 2000). The input data set contained the concentrations and uncertainties of PM$_{2.5}$
mass, S, Cl, K, Ca, Ti, V, Cr, Mn, Fe, Ni, Cu, Zn, Br, Rb, Ba, Pb, EC, OC and LVG for all sampling sites
and seasons. A multisite PMF modelling with 143 samples was performed (Dai et al., 2020). For most
chemical species, all concentrations were higher than the LODs. For some trace elements, the
concentrations were larger than the LODs in >60 % of the samples. The missing data were replaced by
the related median with an uncertainty of 5/6 of the LOD value. The concentrations above LODs were
associated with an equation-based extra standard deviation in accordance with the guidelines of the PMF
manual, which involved the measurement uncertainty, the concentration and the LOD values (EPA,
2017). Elements Cl, Cr, Ni and Rb were specified as weak variables due to their relatively large SDs.

Several test runs were performed with a total number of factors ranging from 3 to 9 to define the base
runs. To explore the goodness of the individual results and to derive robust source apportionment,
additional mathematical tools such as bootstrapping and displacement methods were adopted (Norris et





al., 2014). In bootstrapping, the compliance between the base factors and bootstrapped factors (which
were later selected as the final solution) was >80 %. In addition, the displacement for these solutions did
not show larger changes in the parameter Q and no swap counts of factors occurred.

Multiple linear regression (MLR) modelling was performed to deconvolute the measured $OP_{AA,V}$ and
$OP_{DTT,V}$ data sets as the dependent variables among the main aerosol sources identified by the PMF as the
independent variables (Weber at al., 2021). A linear predictor function was fitted through the dependent
variable points by the weighted least squares (WLS) method. The weights were chosen as the inverse of
the square of the SD for each measured OP. Goodness of the fit was checked by residual analysis. The
significant predictor variables were selected using an *F*-test. The calculations were performed in the
advanced analytics software package Statistica (version 7.1, StatSoft, Germany).

## 3 Results and discussion

### 3.1 Ranges, averages and tendencies

Basic statistics of the daily OP data and atmospheric concentrations obtained from the filters for the whole
sampling interval in the three environments are summarised in Table 1. Some further atmospheric
concentrations measured online and the local meteorological data together with their measuring methods
are given in Sect. S1 in the Supplement. The concentrations of $EC_{FF}$, $OC_{FF}$, $EC_{BB}$, $OC_{BB}$, $OC_{BIO}$ and SOC
derived earlier can be found in previous publications (Salma et al. 2020a, 2022). The present average
concentrations and meteorological data are in line with the results of earlier studies (Salma et al., 2004,
2005, 2020a; Szigeti et al., 2015) suggesting that the overall data set represents typical atmospheric
conditions at the locations. However, several Saharan dust intrusions into the Carpathian Basin happened
in April 2018 (Varga, 2020). The most intensive event reached the region via southern Italy and the
Balkans on 15 April and affected the studied region for a few days.

The average $PM_{2.5}$ mass, $OP_{AA}$ and $OP_{DTT}$ data sets showed three different tendencies with respect to the
locations. This is better visualised with their annual mean and median data in Fig. 1. The averages (i.e.
the medians and means) of the $PM_{2.5}$ mass exhibited a rising trend with levelling off from the rural
background through the suburban area to the city centre. The means of both $OP_{AA,m}$ and $OP_{AA,V}$ data sets
indicated a maximum in the suburban background, whereas the tendencies for their medians were not
fully conclusive. The averages of both $OP_{DTT,m}$ and $OP_{DTT,V}$ data sets showed steadily increasing
behaviour. The differences in the tendencies already suggest that there is a misalignment between the PM



mass and the OP data and that the two assays used show different sensitivity to source types active at the
locations.

**Table 1.** Ranges and medians of oxidative potential (OP) determined by AA and DTT assays and normalised to PM mass ($m$;
$OP_{AA,m}$ and $OP_{DTT,m}$, respectively, in unit of nmol min$^{-1}$ µg$^{-1}$) or to sampled air volume ($V$; $OP_{AA,V}$ and $OP_{DTT,V}$, respectively,
nmol min$^{-1}$ m$^{-3}$), of concentrations for PM$_{2.5}$ mass (µg m$^{-3}$), chemical elements (all in ng m$^{-3}$), elemental carbon (EC), organic
carbon (OC, both in µg m$^{-3}$), levoglucosan (LVG, ng m$^{-3}$) in the rural background, suburban area and city centre.

| Site/ Variable | Rural background | | | Suburban area | | | City centre | | |
|---|---|---|---|---|---|---|---|---|---|
| | Minimum | Median | Maximum | Minimum | Median | Maximum | Minimum | Median | Maximum |
| $OP_{AA,m}$ | 0.01 | 0.12 | 0.28 | 0.02 | 0.14 | 0.33 | 0.03 | 0.11 | 0.23 |
| $OP_{AA,V}$ | 0.1 | 1.5 | 5.2 | 0.3 | 2.4 | 11.7 | 0.3 | 2.1 | 4.9 |
| $OP_{DTT,m}$ | 0.04 | 0.10 | 0.20 | 0.004 | 0.11 | 0.27 | 0.11 | 0.17 | 0.27 |
| $OP_{DTT,V}$ | 0.3 | 1.2 | 3.1 | 0.03 | 1.9 | 6.2 | 1.3 | 2.9 | 6.1 |
| PM$_{2.5}$ | 6 | 14 | 29 | 7 | 18 | 46 | 7 | 18 | 44 |
| S | 51 | 311 | 1043 | 84 | 312 | 823 | 167 | 367 | 952 |
| Cl | 5 | 11 | 28 | 5 | 32 | 118 | 5 | 23 | 71 |
| K | 11 | 56 | 234 | 9 | 65 | 363 | 18 | 91 | 264 |
| Ca | 1 | 33 | 215 | 6 | 73 | 457 | 23 | 104 | 468 |
| Ti | 0.05 | 1.3 | 15 | 0.3 | 1.9 | 26 | 0.6 | 3.1 | 22 |
| V | 0.23 | 0.46 | 1.1 | 0.13 | 0.43 | 1.0 | 0.13 | 0.44 | 1.2 |
| Cr | 0.16 | 0.37 | 0.91 | 0.18 | 0.46 | 1.7 | 0.17 | 0.76 | 7.2 |
| Mn | 0.4 | 2.0 | 16 | 0.1 | 2.1 | 5.6 | 0.5 | 3.3 | 11 |
| Fe | 5 | 32 | 162 | 16 | 63 | 306 | 33 | 102 | 607 |
| Ni | 0.13 | 0.42 | 1.3 | 0.14 | 0.33 | 1.1 | 0.18 | 0.57 | 3.0 |
| Cu | 0.13 | 0.94 | 6.8 | 0.6 | 1.6 | 7.5 | 0.8 | 2.9 | 27 |
| Zn | 0.9 | 7.2 | 40 | 3 | 12 | 53 | 1 | 17 | 48 |
| Br | 0.20 | 0.77 | 3.0 | 0.2 | 1.2 | 4.1 | 0.5 | 1.3 | 2.7 |
| Rb | 0.22 | 0.35 | 0.76 | 0.22 | 0.36 | 0.83 | 0.24 | 0.34 | 0.80 |
| Ba | 1.1 | 2.4 | 12 | 1.1 | 3.1 | 12 | 1.1 | 4.5 | 13 |
| Pb | 0.6 | 3.4 | 28 | 1.5 | 5.3 | 21 | 1.3 | 5.6 | 19 |
| EC | 0.08 | 0.22 | 0.77 | 0.21 | 0.50 | 1.1 | 0.31 | 0.78 | 1.8 |
| OC | 0.9 | 2.3 | 6.0 | 1.0 | 2.9 | 11 | 2.0 | 3.3 | 8.0 |
| LVG | 4 | 38 | 776 | 5 | 106 | 1765 | 8 | 203 | 709 |


Basic statistics of PM$_{2.5}$ mass and OP data separately for each season and the whole year are shown in
Fig. 1. In winter and autumn (the heating period), the OP and PM mass levels were higher than in spring
and summer. This is consistent with the other continental European sites (e.g. Calas et al., 2019; Borlaza
et al., 2022). The heating period-to-non-heating period OP ratios in the urban locations were larger than
for the rural background by factors of ca. 4 for $OP_{AA,V}$ and 1–2 for $OP_{DTT,V}$. There were similar tendencies
in the OP values derived by both AA and DTT assays over the seasons. Except for the $OP_{DTT,m}$ data,
which showed a fairly constant level over the seasons with some higher values in summer, particularly in



the city centre. This can be again linked to the altered chemical composition of PM mass in time and to
the different responses of the two assays to this change.

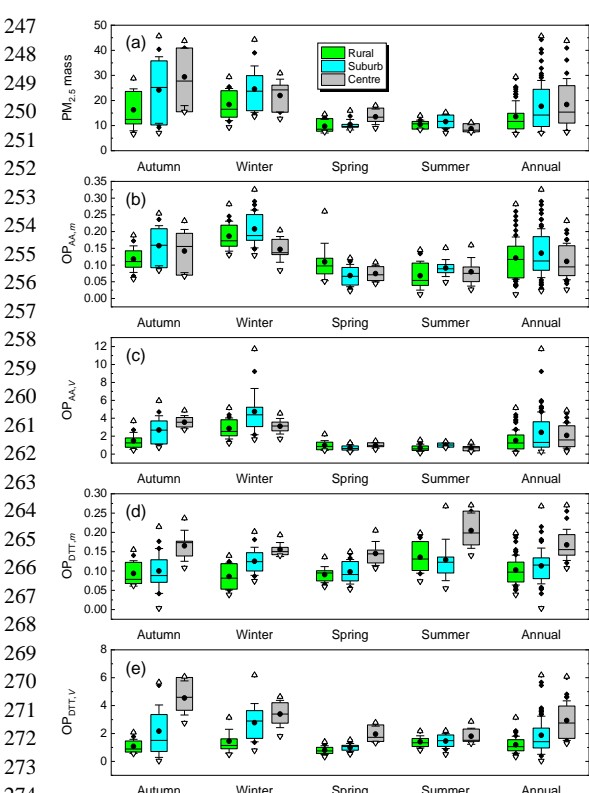

**Figure 1.** Box and whisker plots of PM$_{2.5}$ mass concentration (µg m$^{-3}$; panel a), oxidative potential (OP) determined by AA
and DTT assays and normalised to PM mass ($m$; OP$_{AA,m}$ and OP$_{DTT,m}$, nmol min$^{-1}$ µg$^{-1}$; panels b and d) or to sampled air
volume ($V$; OP$_{AA,V}$ and OP$_{DTT,V}$, nmol min$^{-1}$ m$^{-3}$; panels c and e) in the rural background, suburban area and city centre
separately for each season and over one year. Maximum and minimum values (triangles pointing upward and downward,
respectively), further extreme values (diamonds), the first and third quartiles (lower and upper horizontal borders of the boxes,
respectively), median (horizontal line inside the boxes), means (bullets) and ±1 SDs (whiskers) of the data sets are shown.

There are only a few other OP data sets for the PM$_{2.5}$ size fraction derived by AA and DTT assays. Their
comparison to our OP data is hindered by important experimental details such as the extracted amount of
PM from filters. It can be roughly identified that our median OP values are somewhat larger than those at
other European sites (Daellenbach et al., 2020; Grange et al., 2022, In 't Veld et al., 2023), while they
belong to the middle range of the available results for Japan and China (Kurihara et al., 2022; Yu et al.,





2019). The differences can be also influenced by the exact location type since higher OP data near traffic
sources were observed (Boogaard et al., 2012; Fang et al., 2016; Daellenbach et al., 2020).

**3.2 Consistency between the assays**

The dependencies between the OP data derived by the AA assay and normalised either to $m$ or $V$ on the
corresponding DTT data are displayed in Fig. 2.

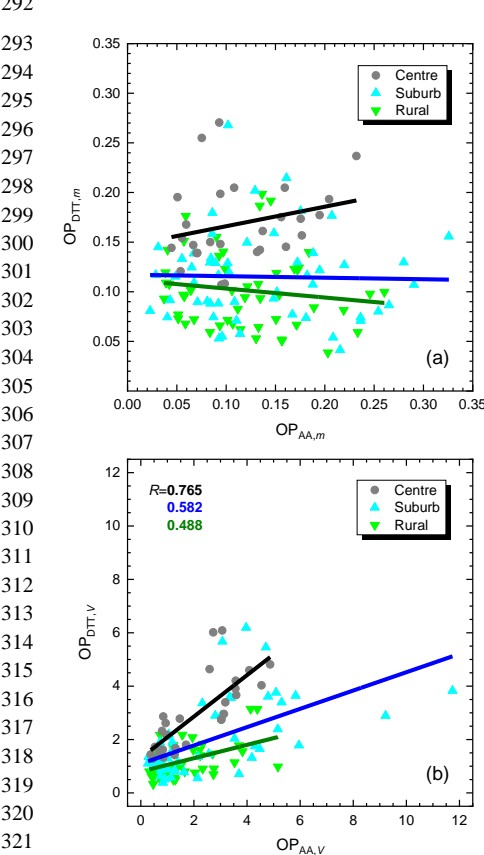

**Figure 2.** Scatter plots of the oxidative potential (OP) values determined by AA and DTT assays and normalised to PM mass

($m$; $OP_{AA,m}$ and $OP_{DTT,m}$, in nmol min$^{-1}$ µg$^{-1}$; panel a) or to sampled air volume ($V$; $OP_{AA,V}$ and $OP_{DTT,V}$, nmol min$^{-1}$ m$^{-3}$; panel
b) separately in the rural background, suburban area and city centre. The coefficients of correlation ($R$s) for the significant
cases are also given.
Pearson's coefficients of correlation ($R$) between the data sets normalised to $m$ (Fig. 2a) were not
significant ($p$=0.05) at the locations. It suggests that the $OP_m$ was controlled by chemical species that
invoked different responses in the assays. However, all correlations between the two OP data sets



normalised to *V* (Fig. 2b) were significant. The slopes with SDs of the regression lines were smaller than
unity (0.25±0.06, 0.34±0.07 and 0.78±0.13, respectively) and increased monotonically from the rural
background through the suburban area to the city centre.

The results suggested that the OP values normalised to sampled air volume were coherent. The AA assay
reacted more sensitively to the changes in chemical composition of PM than the DTT assay at our
locations. The increasing slope of the regression could be also connected to the fact that organics of
biogenic origin exhibit smaller responses in the DTT assays than those of BB (Verma et al., 2015) or of
urban sources in general. The differences could be partly influenced by aerosol photochemical aging and
SOC formation over seasons (Wong et al., 2019; Zhang et al., 2022). More importantly, the conclusions
definitely underline the need for deploying multiple (at least two) OP assays, particularly in cleaner
atmospheric environments, to achieve a more wholistic and consistent picture (Bates et al., 2019; Calas
et al., 2017; Borlaza et al., 2022).

**3.3 Main aerosol sources**

Six factors resolved by the PMF modelling were further evaluated as described in Sect. S2. The following
aerosol sources were identified: biomass burning, suspended dust, road traffic, oil combustion, vehicle
metal wear and mixed industrial source. Similar set of source types was also identified earlier for Budapest
in larger number of samples in winter (Furu et al., 2022).

The apportionments of the $PM_{2.5}$ mass among the main sources are summarised in Fig. 3 separately for
each location and season. The plots reveal that the source contributions changed very substantially over
the year. In winter, BB was the dominant source (with mean contributions of 50 % − 60 %) at all sites. In
autumn, BB and oil combustion were the two most important sources in the rural background with similar
shares (38 %). In the suburban area, BB exhibited very similar (38 %) contribution to the rural
background, whereas oil combustion and the joint importance of road traffic and vehicle metal wear
showed the second largest and similar contributions (20 %). In the city centre, traffic-related sources were
the most important contributors (40 %). In spring, oil combustion prevailed (60 %) in the rural
background. Its contribution monotonically decreased through the suburban area (46 %) to the city centre
(26 %). In parallel with this tendency, the joint share from road traffic and vehicle metal wear increased
monotonically (from 17 % through 30 % to 49 %) in the same order of the locations. The contributions
from suspended dust in spring were also significant at all locations accounting for approximately 15 %.
They were influenced by the Saharan dust intrusion episodes extending over the whole Carpathian Basin
in this season. In summer, oil combustion was again the dominant source (66 %) in the rural background





and showed a decreasing share for the suburban area (45 %) to the city centre (41 %). Contrary, the effects
of road traffic monotonically rose (from 13 % through 31 % to 44 %). This increasing tendency was
preserved in the other seasons as well. The unaccounted sources and their possible effects on the final
results are discussed in Sect. 3.6.

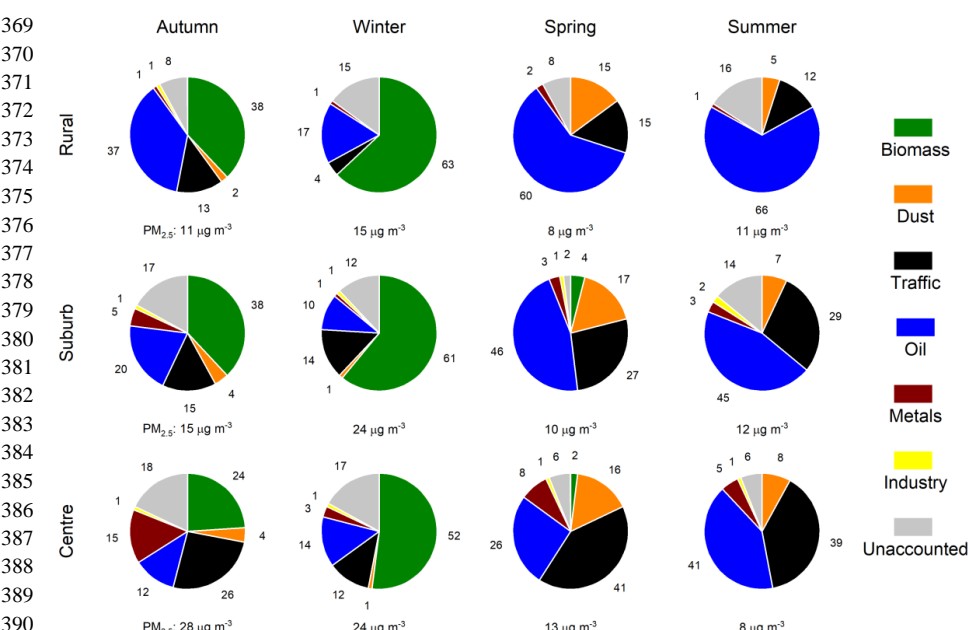

**Figure 3.** Mean contributions of biomass burning, suspended dust, road traffic, oil combustion, vehicle metal wear, mixed
industrial source and unaccounted sources to the atmospheric concentration of $PM_{2.5}$ mass (in %) as derived by the PMF
modelling in the rural background, suburban area and city centre in different seasons. The median atmospheric concentrations
are shown under the circle charts.

The apportionments of Cu and Fe, which are of special interest for OP, among the main aerosol sources
as derived by the PMF modelling are shown in Figs. S7 and S8. Copper mainly or dominantly originated
from motor vehicles, i.e. vehicle metal wear and road traffic sources except for the rural background in
winter and autumn. The outstanding role of road vehicles is confirmed by our earlier results for a street
canyon in central Budapest (Salma and Maenhaut, 2006). The smallest shares from vehicles occurred in
winter (22 %, 39 % and 65 %, respectively in the rural background, suburban area and city centre), while
the maximum contributions happened in summer (51 %, 65 % and 87 %, respectively). The contribution
of unaccounted sources in the rural background in winter was large (33 %), which could modify the
apportionments. The role of BB in Cu emissions could be possible explained by illegal industrial and
household waste burnings together with biomass (Sect. S2; Hoffer et al., 2021).




In the city centre, the vehicle sources showed the largest contributions to Fe (53 % − 74 %) in all seasons,
and dust was its second most intensive source (30 %−36 %) in spring and summer. At the other two
locations, Fe in spring was unambiguously dominated by dust (ca. 55 %), which was influenced by the
Saharan dust intrusion. Suspended dust remained the most important source in the rural background in
summer, whereas it became comparable to the traffic-related sources in the suburban area. Vehicles
tended to be the second largest Fe source (26 % − 45 %) in the rural background and suburban area. Their
contributions could be partly also associated with the resuspended road dust generated by moving
vehicles. In autumn, the shares in the rural background were more or less balanced among the main
sources, while the vehicle contributions were increased in the suburban area.

The examples of Cu and Fe demonstrated broadly varying spatial and temporal tendencies in the source
contributions of OP-active chemical species, and point to the potentials of regulatory measures based on
specifically selected source types.
**3.4 Oxidative potential and aerosol sources**
The OP data normalised to sampled air volume were apportioned to the main aerosol sources, i.e. of BB,
suspended dust, road traffic, oil combustion, vehicle metal wear and mixed industrial source using the
MLR method with the WLS approach. The slopes and intersects of the regression lines calculated for the
whole data set at each sampling location are summarised in Table S4. In a few cases, negative slopes were
obtained. This is commonly found with this approach, but the absolute values of the negative slopes
should be relatively small. This was not the case for the vehicle metal wear and $OP_{DTT,V}$ pair in the rural
background, for the road traffic and $OP_{AA,V}$ pair in the suburban area, and for the oil combustion and
$OP_{AA,V}$ pair in the city centre. The intersects of the $OP_{DTT,V}$ in the suburban area and city centre also
resulted in statistically nonzero values. These cases jointly indicate that there could be some aerosol
sources missing in the PMF modelling due probably to the unavailability of some important marker
variables and to the limited number of samples. The shortcoming is further discussed in Sect. 3.6. It cannot
be excluded that this imperfection influences the order and mainly the contributions of the OP sources.
To improve the attribution of the OP to the identified aerosol sources, the MLR model with the WLS
approach was also performed with forced positive slopes option. Its constrained results are summarised
in Table 2.

With this latter option, all intersects became statistically insignificant ($p<0.05$) from zero. The AA assay
yielded significant slopes with BB, road traffic and oil combustion in the rural background, with BB and



suspended dust in the suburban area and with BB, and road traffic in the city centre. The DTT assay
resulted in significant slopes with BB and road traffic, with BB and oil combustions and with BB and
road traffic in the three environments. Comparing the fitted MLR parameters obtained by the constrained
and non-constrained WLS approaches (Tables 2 and S4) shows that the orders of the sources did not
change substantially, and that the positive slopes obtained by the two models are comparable. At the same
time, the importance of oil combustion decreased in some occasions. These likely indicate that the derived
ranks of the OP sources are sensible approximations to reality with some larger uncertainties of their
contributions.

**Table 2.** Slopes and intersects of the multiple linear regression with the weighted least squares approach and forced positive
slopes option between oxidative potential (OP) determined by AA and DTT assays and normalised to sampled air volume
($OP_{AA,V}$ and $OP_{DTT,V}$, respectively) and the main aerosol sources of biomass burning, suspended dust, road traffic, oil
combustion, vehicle metal wear and mixed industrial source derived by PMF modelling in the rural background, suburban area
and city centre. The number of samples available ($n$) and the adjusted coefficients of determination ($R^2$) are also shown.
Nonsignificant values are in *Italic* font.

| Location/ | Rural background | | Suburban area | | City centre | |
|---|---|---|---|---|---|---|
| Main source | $OP_{AA,V}$ | $OP_{DTT,V}$ | $OP_{AA,V}$ | $OP_{DTT,V}$ | $OP_{AA,V}$ | $OP_{DTT,V}$ |
| Biomass burning | 1.414 | 0.873 | 0.792 | 0.622 | 1.073 | 0.788 |
| Suspended dust | *0.113* | – | 0.569 | *0.018* | *0.025* | *0.090* |
| Road traffic | 1.010 | 0.959 | – | *0.181* | 0.357 | 0.887 |
| Oil combustion | 0.279 | – | *0.522* | 0.968 | – | *0.488* |
| Vehicle metal wear | *0.056* | – | – | – | *0.018* | *0.091* |
| Mixed industrial | – | *0.085* | 0.172 | 0.086 | *0.142* | – |
| Intersect | *–0.160* | *0.358* | *–0.473* | *–0.497* | *–0.081* | *–0.362* |
| $N$ | 52 | 51 | 56 | 55 | 28 | 28 |
| $R^2$ | 0.974 | 0.877 | 0.717 | 0.779 | 0.858 | 0.811 |


The driving effect of BB on OP has been highlighted in other studies (e.g. Verma et al., 2015; Lionetto et
al., 2021; Borlaza et al., 2022). The intensity of BB in the Carpathian Basin is, however, large only in the
heating period (autumn and winter), and much lower outside this interval. To refine the apportionment of
the OP data to aerosol sources active in the non-heating seasons, the MLR modelling was repeated with
the joint data set of all sites split into heating and non-heating periods. These results confirmed that BB
shows overwhelming contributions to the OP values in the heating period. More importantly, the obtained
results also imply that the shares from vehicles (i.e. joint sources of road traffic and vehicle metal wear)
in the non-heating period to OP prevailed. This is in line with the attributions of some transition metals



such as Cu and Fe to these aerosol sources (Figs. S7–S8 and Salma and Maenhaut, 2006), and also points
to the remarkable role of primary traffic emissions in causing oxidative stress in spring and summer.

Secondary organic aerosol under anthropogenically influenced conditions was proven to be one of the top
factors for OP (Srivastava et al., 2018; Wong et al., 2019; Daellenbach et al., 2020; Borlaza et al., 2021a,
2021b; Pye et al., 2021; Zhang et al., 2022). The involvement of the SOC concentrations into the PMF
was hampered by their smaller count and larger relative uncertainty (up to 30 % – 50 %; Salma et al.,
2022). Instead, we investigated the correlations between the OP data sets and SOC concentrations or
SOC/OC ratios. The dependencies of the $OP_{DTT,V}$ on the SOC are shown in Fig. S9. The OP values at the
urban locations tended to increase with the SOC in parallel with each other, while the OP was rising in a
smaller rate in the regional background. The reasons behind these observations likely include the distinct
effects of biogenic and anthropogenic secondary organic aerosols typically present in different portions
at the sampling locations. The results may also indirectly indicate that photochemical aging processes
impact the toxicity of PM as well (Kodros et al., 2020). There were no significant correlations obtained
for the other data pairs.

**3.5 Oxidative potential and air quality**

Particulate matter mass was proven to be the most important proximity metric from the criteria air
pollutants for the general air quality in the Carpathian Basin (Salma et al., 2020a, 2020b). Therefore, the
relationships between the $PM_{2.5}$ mass and OP data sets normalised to sampled air volume were separately
investigated. Their correlation dependencies were all significant (Fig. 4). Spatial and temporal
correlations between these variables from low to moderate were also observed in earlier studies under
broadly varying conditions (Künzli et al., 2006; Boogaard et al., 2012; Yang et al., 2015; Fang et al.,
2016; Chirizzi et al., 2017).

The dependencies for the $OP_{DTT,V}$ (Fig. 4a) resulted in two almost parallel lines (with slopes and SDs of
$0.11\pm0.01$ and $0.13\pm0.01$ nmol min$^{-1}$ µg$^{-1}$, respectively) in the city centre and suburban area, while a
smaller slope ($0.051\pm0.012$ nmol min$^{-1}$ µg$^{-1}$) was observed in the rural background. The situation for the
$OP_{AA,V}$ (Fig. 4b) was less obvious but somewhat similar to $OP_{DTT,V}$. The regression lines for the rural
background and suburban area tended to be fairly parallel with each other (with slopes and SDs of
$0.16\pm0.02$ and $0.18\pm0.02$ nmol min$^{-1}$ µg$^{-1}$, respectively), whereas the slope for the city centre was smaller
($0.096\pm0.019$ nmol min$^{-1}$ µg$^{-1}$). The intersects could be typically regarded to be zero within the
uncertainty interval.





The parallel tendencies may indicate that the effects of the PM chemical compositions on the given assay were close to each other at the sampling locations with the parallel lines. This was likely caused by spatial and temporal similarities in the main sources such as road traffic and resuspended dust particularly for the DTT assay, and biomass burning especially for the AA assay (Salma et al., 2020a). Particles in the third environment (with the smaller slope) likely contained less OP-active components from the point of the given assay and, therefore, the increase in the OP was more modest. This interpretation is confirmed by earlier similar conclusions (Daellenbach et al., 2020). Nevertheless, it should be stressed that all slopes were substantially and much smaller than unity. This implies that the air quality regulatory measures based on the $PM_{2.5}$ mass are expected to result in smaller improvements in oxidative stress compared to dedicated measures that specifically target (appropriately selected) aerosol sources.

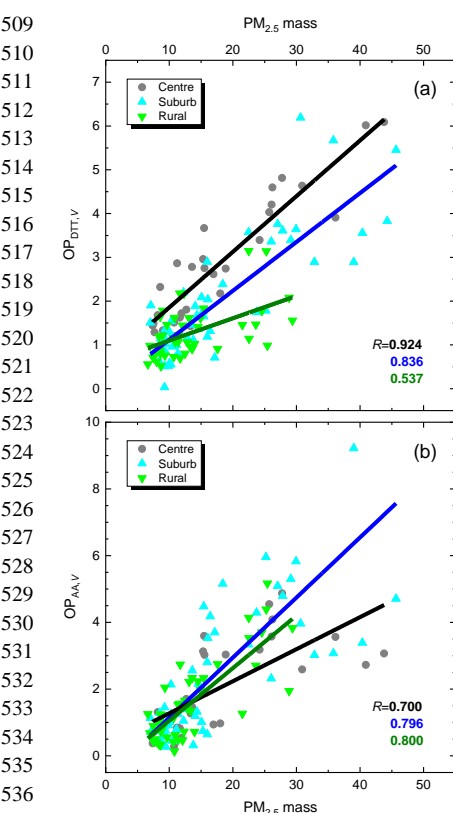

**Figure 4.** Scatter plots of the oxidative potential (OP) determined by DTT (a) and AA (b) assays and normalised to sampled air volume ($V$; $OP_{DTT,V}$ and $OP_{AA,V}$, respectively, in nmol $min^{-1}$ $m^{-3}$) and $PM_{2.5}$ mass concentrations ($\mu g$ $m^{-3}$) for the rural background, suburban area and city centre. The lines represent linear regressions of the data sets. The coefficients of correlation ($R$s) are also indicated.



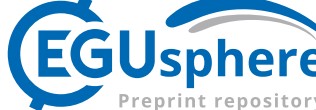

**3.6 Limitations and later possibilities**

The total numbers of the samples collected at each location represent a limitation mainly for the PMF modelling. To overcome this problem, we applied multisite PMF. It was implicitly assumed in this approach that the main sources active at the locations can be characterised by similar chemical profiles. This is not completely fulfilled for all seasons. An example is the suspended dust which is virtually fugitive mineral and soil dust made of geogenic elements in the rural backgrounds. In the urban sites, it contains further constituents originally generated by anthropogenic activities, which settled down to surfaces and later entered into the air again by resuspension. It is mentioned that the PMF modelling on the separate locations yielded fairly similar results to the multisite approach, while the statistical uncertainties of these latter calculations were favourable.

The unavailability of some secondary inorganics, mainly nitrate and ammonium ions in the present analytical data sets introduced another limitation. Their contributions were likely contained in the unaccounted sources of the PMF modelling. They ranged up to 18 % and showed contributions mainly in colder (heating) seasons or in summer for the rural background and suburban area. Fortunately, pure secondary inorganic constituents are associated with lower contribution to PM toxicity (Cassee et al., 2013; Daellenbach et al., 2020), although they can influence the OP through acid mediated dissolution of transition metals (Fang et al., 2017). However, the robustness of the PMF modelling can influence the final apportion of the OP among the resolved sources.

Larger number of samples and extended list of variables are pronouncedly required because of the basin character of the region of interest. The poorest air quality in the whole Carpathian Basin generally occurs in winter (Salma et al., 2022), when persistent anticyclonic weather situations and lasting temperature inversions happen for longer times. During these intervals, the time series of aerosol constituents, even of different origins, change coherently at many locations due to the common effects of regional meteorology. This dependency can further encumber the separation of the aerosol sources in PMF modelling (Salma et al., 2004).

The present results and conclusions can be further improved by involving additional important chemical species and markers, mainly water-soluble metal ions, water-soluble OC, primary biogenic OC and PAHs. Extended research is required to address some additional relevant sources such as coal combustion, biogenic emissions and illegal waste burning. Investigations of size-fractionated aerosol samples with several toxicity indicators including intracellular tests and possible synergism or antagonism among



chemical species could bring further insights into the oxidative stress research. The experience gained in the present work, which was conducted in a systematic manner for the first time in this region, can form valuable experience in planning further related studies.

## 4 Conclusions

We showed that the OP induced by $PM_{2.5}$ mass and determined by the AA and DTT assays in the rural (regional) background of the Carpathian Basin, in the suburban area and centre of its largest city of Budapest differed substantially and in a complex manner with location and changed considerably and consistently with season. The alterations were mainly caused by varying intensities of the main aerosol sources and possibly by other specific seasonal features. Biomass burning clearly exhibited the dominant influence at all locations in the heating period. Several pieces of indirect evidence suggest that the joint effects of motor vehicles involving road traffic and vehicle metal wear played the most important role in summer and spring, with considerable contributions from oil combustion and resuspended dust.

The most severe daily PM health limit exceedances in Budapest (and several other European cities) occur in winter due to both residential heating and meteorological effects. The contributions of BB to OP are the largest during this season. Thus, human exposure to high pollution levels are further exacerbated by the chemical composition which causes increased oxidative stress. As far as the sources related to motor vehicles are concerned, large traffic intensities frequently occur in city centres, which generate dangerous hotspots of particularly OP-active species. In these sites, an enhanced exposure of public in summer and spring often coincides with high population densities.

Our conclusions imply that targeting the PM mass in general does not efficiently reduce the oxidative burden from PM exposure. Instead, substantial health improvements could be achieved by focusing on some specific source types such as BB in winter and vehicle traffic in non-heating period. The former source may have timely consequences since it is expected to be increased in the near future. The non-exhaust emissions from vehicle traffic are anticipated to gain in relevance as well since high-efficiency exhaust gas aftertreatment devices had been already adopted to internal combustion engines and because of global spreading of electric vehicles. The advantages of BB and electric cars are often emphasized, while their potential drawbacks have been less disseminated. It is needed to further investigate their distinctive health effects for setting up effective mitigation policies and season-specific regulations.

*Data availability.* The observational data are available from the corresponding author.

*Supplement.* The supplement related to this article is available online at: *to be completed.*



*Author contributions.* MV evaluated the data, performed the modelling calculations, prepared figures, participated in
interpreting the results and contributed to writing the manuscript; GU and J-LJ managed the OP measurements, GU, J-LJ and
PD participated in interpreting the results and revising the manuscript; ZsK and EP manged the PIXE measurements and
participated in interpreting the PMF results; IS conceived the research, arranged the sample collections, evaluated and
interpreted the results, prepared figures, wrote the manuscript. All coauthors reviewed and commented on the manuscript.
*Competing interests.* The authors declare that they have no conflict of interest.
*Acknowledgements.* The authors are grateful to Anikó Angyal of the Institute for Nuclear Research for the PIXE measurements,
to Attila Machon of the Hungarian Meteorological Service for the assistance in the sample collections and the OC/EC
measurements, and to Anikó Vasanits of the Eötvös Loránd University for the LVG measurements.
*Financial support.* This research has been supported by the Hungarian Research, Development and Innovation Office (grants
K132254 and K146915), the European Regional Development Fund and the Hungarian Government (grant GINOP-2.3.3-15-
2016-00005) and the New National Excellence Program of the Ministry for Innovation and Technology from the source of the
Hungarian Research, Development and Innovation Fund (ÚNKP-21-3). The OP analysis was supported by the ACME program
(ANR-15-IDEX-02) and ANR Get OP Stand OP program (ANR-19-CE34-0002-01), and were analyzed at the Air-O-Sol
facility at IGE, made possible with the funding of some laboratory equipment by the Labex OSUG@2020 (ANR10 LABX56).

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
