# Peer review of "Oxidative potential in rural, suburban and city centre atmospheric environments in Central Europe 2"

_EGUsphere, 2023_

## Author Comment (AC1)

**Response to Referee number 1**

25 August 2023

The authors would like to thank Referee no. 1 very much for her/his expert, detailed and valuable comments to further improve and clarify the MS. We have considered all recommendations and made the appropriate alterations. Our specific responses are as follows, while the textual modifications amended to the MS can be traced in its marked-up version, which is available online.

This MS presents an assessment of aerosol oxidative potential in a number of environments in central Europe. The MS is well written and follows a straightforward structure. It presents a comprehensive analysis covering from the quantification of major and trace aerosol components to source contribution with PMF and finally apportioning the relationship between sources and OP results. The OP results are interpreted in terms of seasonality and links with emission sources in the study area. This is an interesting study which in principle merits publication. The main issue in my view is the methodology applied (acellular assays, AA and DTT), their lack of comparability and different responses obtained across studies. While this is simply the current state of the art, the authors should expand on these limitations, what the advantages with regard to other current methods are and how these limitations impact their results.

Response 1: We complemented the MS with more detailed discussions on the difficulties of the comparability of the OP determination methods in general and added some potentials and methodological advances required in the near future.

**Specific comments:**

- lines 40-52, please add references and/or criteria for the selection of these 8 factors

Response 2: The listed eight factors cover the major properties involved in the pathogenesis of the diseases in general. A review-type reference was added to back them.

- lines 71 and below: the authors discuss the increasing interest in methods to determine OP using in vivo, in vitro cellular and in vitro acellular assays, but then they move on to only discuss the acellular assays (lines 75 and below) by stating that "The OP is frequently measured by acellular assays for exogenous ROS". At least one paragraph should be dedicated to discussing the limitations/advantages of the acellular vs cellular (in vitro or in vivo) methods, and their implications regarding the results presented in this work. Some (non exhaustive) examples of authors working on cellular methods are

Janssen, N. A. H., Yang, A., Strak, M., Steenhof, M., Hellack, B., Gerlofs-Nijland, M. E., Kuhlbusch, T., Kelly, F., Harrison, R., Brunekreef, B., Hoek, G., & Cassee, F. (2014). Oxidative potential of particulate matter collected at sites with different source characteristics. Science of the Total Environment, 472, 572–581. https://doi.org/10.1016/j.scitotenv.2013.11.099

Gerlofs-Nijland, M. E., Bokkers, B. G. H., Sachse, H., Reijnders, J. J. E., Gustafsson, M., Boere, A. J. F., Fokkens, P. F. H., Leseman, D. L. A. C., Augsburg, K., & Cassee, F. R. (2019). Inhalation toxicity profiles of particulate matter: a comparison between brake wear with other sources of emission. Inhalation Toxicology, 31(3), 89–98. https://doi.org/10.1080/08958378.2019.1606365

Bessa, M. J., Brandão, F., Fokkens, P., Cassee, F. R., Salmatonidis, A., Viana, M., Vulpoi, A., Simon, S., Monfort, E., Teixeira, J. P., & Fraga, S. (2020). Toxicity assessment of industrial engineered and airborne process-generated nanoparticles in a 3D human airway epithelial in vitro model. Nanotoxicology, 15(4):542-.

Stone, V., Miller, M. R., Clift, M. J. D., Elder, A., Mills, N. L., Møller, P., Schins, R. P. F., Vogel, U., Kreyling, W. G., Jensen, K. A., Kuhlbusch, T. A. J., Schwarze, P. E., Hoet, P., Pietroiusti, A., Vizcaya-Ruiz, A. de, Baeza-Squiban, A., Teixeira, J. P., Tran, C. L., & Cassee, F. R. (2017). Nanomaterials Versus Ambient Ultrafine Particles: An Opportunity to Exchange Toxicology Knowledge. Environmental Health Perspectives, 125(10).

Response 3: The presentation of the cellular in vitro and in vivo methods and their comparison to acellular assays are beyond the scope of the present MS. Cellular methods were not used in the present work. The comparison could be performed in a dedicated (international) study. We extended the references to cellular methods as requested which direct the interested readers and added a note on the complexity of the various OP metrics and the whole field.

- line 98: "It is important to extend the studies on this emerging health-related metric", wouldn't it be necessary first to agree on a comparable method? The authors have just discussed the lack of comparability between the AA and DTT methods, and concluded that "they exhibit different responses to various groups of ROS-generating compounds and their bioavailability". What can be concluded from this? What is the point of extending results to other regions if they cannot be compared? Please discuss how the authors plan to approach this.

Response 4: The measuring methods for the oxidative stress and OP develop in parallel and in close association. Extending the research on OP to further atmospheric environments and comparing the results in a method-wise manner can definitely contribute both to creating better insights into the field and to creating new ideas on more representative measuring approaches. It cannot be expected for the time of being that one experimental procedure of the possible realisations is selected as a standard method – if it will be possible at all. Different methods are sensitive in different way to various aerosol components. The results of a particular method can be compared to those obtained by the identical procedure for different locations or seasons. We briefly completed the text with these aspects to clarify our motivation.

- line 121, how many samples were collected per location and per season? The numbers described here are relatively low to apply PMF. If all the samples pooled together for the PMF analysis please discuss the limitations, e.g., different emission profiles in rural vs city sites. Line 186 confirms that a multisite approach was applied. Please discuss. Section 3.6 discusses these limitations; it might be more useful for the reader if the discussions on

limitations are distributed and found in the respective sections they refer to. For example, lines 543-551 could be moved to the paragraph containing line 121.

Response 5: The samplings including locations, time intervals and number of filters were described in section 2.1 Sample collections. Further details were already given in a previous paper, which was cited. A multisite PMF source apportionment was performed on the joint data set as correctly recognised by the Referee. The total number of samples (143) was given in section 2.4 Mathematical models. We clarified all these better. The limitations of the multisite PMF were discussed in a separate section 3.6 Limitations and later possibilities together with the other similar effects. Another plausible possibility would be to distribute the individual aspects of this treatment into the related sections. We think that presenting a joint, more holistic overview on all limitations at one place prevails over splitting them into parts because they exert their effect on the final results jointly. For this main reason, we would prefer keeping the present structure of the MS.

- line 148, please reduce the number of self-citations; e.g., surely an earlier reference can be provided for the EC tracer method

Response 6: The present MS fits into a series of papers on carbonaceous aerosol components in the Carpathian Basin, their source types and finally, on the oxidative potential of the sources. The cited articles by the present coauthors not only refer to the utilised methodology for the aerosol constituents derived earlier such as SOC, but they also contain the related factual atmospheric concentrations at the identical site and in the same samples. Nevertheless, we reduce the number of self-citations as requested. Six self-citations do not seem to be an unusually large number.

- line 158, the OP of the extracts was measured without filtration, how can the authors be sure their results did not suffer from interference from quartz fibres extracted unintentionally from the filters by vortex agitation? This is a known artefact linked to the toxicity of quartz materials, especially fibrous materials.

Response 7: The AA and DTT acellular assays are based on the redox reactions of specific chemical compounds, i.e. ascorbic acid and dithiothreitol with aerosol chemical species. Trace amounts of quartz are not expected to considerably alter the chemical reaction rates in contrast to causing toxicity in living biological cells as fibres. In addition, the filtration could disadvantageously influence the overall impact of PM by removing undissolved components

with possible active surface area (such as soot particles) and by altering the chemical equilibrium between the dissolved and undissolved chemical species.

- line 227, "steadily increasing towards the city centre", as in the case of PM2.5? Then the OPDDT and PM2.5 results would be aligned, but this doesn't seem evident from Figure 1. Please clarify.

Response 8: The word "averages" in the sentence "The averages of both $OP_{DTT,m}$ and $OP_{DTT,v}$ data sets showed steadily increasing behaviour." remained in the text by a mistake when adopting the changes of all coauthors. It should have been replaced by the "mean". After the present comment, we decided to remove the sentence to increase the clarity of the text.

- line 282, the authors state "Their (AA and DTT results) comparison to our OP data is hindered by important experimental details such as the extracted amount of PM from filters". Aren't the OP results normalised by PM mass? An in depth assessment of the uncertainties of the OP results should be presented, taking into account that the amount of PM extracted is another source of uncertainty. Line 284, "It can be roughly identified that our median OP values are somewhat larger", please remove the subjective terms (roughly, somewhat larger…". It is unclear whether the comparisons reported are reliable, based on the authors' previous statements regarding comparability.

Response 9: Our OP data were obtained in sample extracts with a PM mass concentration of $10 \ \mu g \ ml^{-1}$, whereas some other authors use $25 \ \mu g \ ml^{-1}$, while some further authors do not use isoconcentration extracts at all. All choices have arguments for and against. Our intention here was to mention that the different sample preparations (even of the same filters) may influence the strict comparability of the results. This was expressed by the word "roughly". We modified and extended the sentence to communicate our purpose better and deleted the words roughly and somewhat. The raised aspect of the sample extraction method as a new component of the overall uncertainty was added to section 3.6 Limitations and later possibilities. At the same time, the comparisons of the results obtained by an identical sample preparation and measurement method for different environmental types are sensible, while comparing the data derived by different experimental methods applied for similar sample types can reveal varying sensitivity of these methods to various chemical components and sources. We added these aspects and removed the subjective term from the sentence mentioned.

- line 341, another example of subjective terminology "definitely underline", please remove. Line 342, "wholistic" should be "holistic"

Response 10: Both requests were adopted.

- lines 345-348, please elaborate on the oil source, what is the authors' interpretation, precisely? This source accounts for almost 70% of PM mass in summer and spring in the rural area, which is somewhat surprising. Unless the site is located in close proximity to an industrial activity (in which case the site should be renamed), it is likely that the source refers to long-range transported secondary aerosols, as opposed to direct oil combustion. This would also be consistent with the decreasing relevance of this source towards the city centre, where contributions from primary sources (e.g., traffic) are higher in relative terms. Please review.

Response 11: This source was identified as oil combustion mainly on the presence and amounts of their marker chemical species and on the tendencies in the time series of the factor intensity. We also noted that it was likely and partly mixed with coal combustion as well due to the collinearity of these sources caused by the seasonal trends and meteorological conditions in winter. It cannot be excluded due to the unavailability of the secondary inorganics that the source is also associated with long-range transport particularly in the rural background and nonheating period. This could be also related to limitations of the multisite approach. The primary role of the oil combustion is indicated by the relatively large contributions to the PM$_{2.5}$ mass. This would be less for the long-range transported secondary aerosol particles alone at the urban sites (Salma et al., Elemental and organic carbon in urban canyon and background environments in Budapest, Hungary, Atmos. Environ., 38, 27–36, 2004). We added this additional mixing to the interpretation at several places in the MS and Supplement.

- lines 464 and below: the authors state that "shares from vehicles (i.e. joint sources of road traffic and vehicle metal wear) in the non-heating period to OP prevailed", and that "points to the remarkable role of primary traffic emissions in causing oxidative stress in spring and summer", why only in the non-heating season? If OP is driven by particle chemical composition (I.e., sources) then the effect of traffic aerosols should also be present during the winter months, even if the impact of BB aerosols on OP is even larger during the winter months. Is this the case? Can the impact of BB and traffic on OP be effectively disaggregated, or is there a compound effect?

Response 12: The effect of BB on the OP was overwhelming at all three locations in the heating period independently of the intensity of the vehicle road traffic. The latter changed substantially among the rural background and urban sites. The sentence was modified the express this more precisely. We fully agree with the Referee that the possible synergism or antagonism between BB and vehicle traffic is an important challenging issue, which should be investigated and clarified in further dedicated studies. Section 3.6 Limitations and later possibilities was extended briefly to involve these questions.

- line 481, please define "proximity metric"

Response 13: The sentence was changed to avoid this expression as: Particulate matter mass was proven to be the most important component from the criteria air pollutants in the Carpathian Basin (Salma et al., 2020a, 2020b). Generally, this measure expresses the air quality.

Imre Salma
corresponding author

---

## Author Comment (AC2)

**Response to Referee number 2**

25 August 2023

The authors would like to thank Referee no. 2 very much for his/her expert, detailed and valuable comments to further improve and clarify the MS. We have considered all recommendations and made the appropriate alterations. Our specific responses are as follows, while the textual modifications amended to the MS can be traced in its marked-up version, which is available online.

The OP of three regions in and around Budapest; urban, suburban and rural, over all four seasons is investigated in this paper. The DTT and AA assays are used to measure OP, and source apportionment and linear regression employed to determine the sources affecting these assays at the different locations and for different seasons. The method has been widely applied in other European locations by some of the co-authors, so although the methods are not new, the results are since the location is novel. Overall, the results are interesting, if OP can be/is linked to health endpoints, since they increase the knowledge of factors affecting PM2.5 OP, which are consistently showing the importance of biomass burning and non-tailpipe vehicle emissions. The analysis is very detailed and for the most part the paper is well organized and clearly written. This paper covers an important topic and is suitable for publication in ACP with the following revisions to consider.

Last line of Abstract, does this imply use of OP or just sources in developing regulations?

Response 1: Utilisation of OP for expressing the health endpoints caused by PM would/could be the final goal from several aspects. For the moment, regulations of some source types seem more feasible. The sentence was modified to express these main options.

Line 40, how are ambient particles biologically complex? Is this referring to point 8) in the following lines? Please clarify.

Response 2: By referring to biological complexity of aerosol particles, we meant the presence of active substances of biological origin such as bacteria, viruses, pollens and moulds. This is later expressed in point 8 (section 1 Introduction and objectives).

Lines 66 to 68, this is an incomplete list of current studies linking OP to health effects. Eg, the Weichenthal group has published a number of papers on this topic in the last few years. There may be more, but that could be a place to start an expanded literature search. A better literature review is needed here since this is a critical argument on why these assays are useful, ie, that there is a link to health effects.

Response 3: The MS already contains many citations. We further extended the list of references as requested and emphasized the existing evidence between the $OP_{AA}$ and $OP_{DTT}$ on the one side and the concrete health effects on the other side.

Lines 128 to 130, how was water on the filters controlled or not controlled as part of the mass measurement (ie, equilibrium reached as some low RH)?

Response 4: The exposed and blank filters were pre-equilibrated before weighing them twice in each case at a room $T$ of 19–21°C and RH of 45 %–50 % for at least 48 h. The details of the usual analytical methods were described in a previous article, which reference was added now.

Line 139, if metal ions, such as Cu(II), Fe(II), Mn(II) play a role in OP (see line 92), why were total metals analyzed. How does this influence the interpretation of the data, ie, how can the metals data be linked to OP? One might consider the solubility of the metals. Low soluble metals, such as iron may have little relation to the total iron concentration, whereas for copper with higher solubility, the issue may be different.

Response 5: The transition and heavy metals were determined by PIXE analysis, which gives their total concentration. Their dissolution actually happens in human respiratory tract lining fluid, and this is expected to be larger than in water. Furthermore, an acid mediated dissolution of transition metals (Fang et al., Highly acidic ambient particles, soluble metals, and oxidative potential: a link between sulfate and aerosol toxicity, Environ. Sci. Technol., 51, 2611–2620, 2017) may further increase the solubility. The lung bioaccessibility was assessed in the present work by the total amounts of the chemical species as the first approximation. A short sentence was added to specify this.

Line 158, are insoluble species included in the analysis? Specifically, what fraction of insoluble species are expected to be included in the measured OP. Does soot/EC, which may have surface active species, contribute? More quantitative details are needed here.

Response 6: The sample extracts were not filtered. All insoluble chemical species including those with active surface area were involved in the analysis. It is a good, but rather challenging idea to estimate the fraction of insoluble species in the suspension. A brief comment and a reference were included in the text on this.

Line 331, how are the intercepts on the DTT vs AA in both plots of Fig 2 explained?

Response 7: The intercepts of the regression lines in Fig. 2a and 2b can be related to and can indicate the minimal response of the different methods to the sample types or some specific components. Their interpretation, particularly in Fig. 2a, is not straightforward and definitely needs additional investigations. We would like to avoid drawing specific conclusions here.

Lines 335 to 343, the paragraph is not very clear. Why are the assays coherent based on both the mass and volume normalized data? Also, I assume the statement that AA is more sensitive than DTT is because the slope of DTT vs AA is less than 1 (might state this explicitly). But what if DTT is just more sensitive to a more chemical species? Reference to sources here (SOC, BB) seems out of place, it has not been discussed yet.

Response 8: The statement on the coherency was confined only to the OP normalised to sampled volume (line 335 of the original MS). By this, we intended to express the significant correlation between the AA and DTT methods. The section was reformulated to be more specific and to increase its clarity. The discussions related to BB and SOC sources were replaced to section 3.4 Oxidative potential and aerosol sources.

Source apportionment does not show any secondary biogenic SOA? Also, can PMF be performed here for a given location and season given the limited number of samples?

Response 9: Separate biogenic SOA factor did not appear in the source apportionment since the available SOC concentrations could not be unfortunately included in the calculations due to their smaller count and larger (up to 30 % – 50 %) relative uncertainty. The PMF modelling for a given location and separate seasons was not attempted because of insufficient number of samples for these specific cases. Nevertheless, we utilised the SOA and biogenic OC as supporting data mainly in the regression analyses. These two quantities were derived earlier by an EC tracer method and a coupled radiocarbon-levoglucosan marker method, respectively from the same samples (Salma et al., Fossil fuel combustion, biomass burning and biogenic sources of fine carbonaceous aerosol in the Carpathian Basin, Atmos. Chem. Physics, 20, 4295–4312, 2020; Salma et al.: Secondary organic carbon and its contributions in different atmospheric environments of a continental region and seasons, Atmos. Res., 278, 106360, 2022). Additional PMF test runs to the base case were discussed in sections 2.4 Mathematical models and 3.6 Limitations and later possibilities, which were briefly extended.

Line 424 and throughout. Is intersect the proper term for the regression intercept? Maybe either is acceptable?

Response 10: The term intersect was replaced by intercept in all cases.

Discussion of slopes of DTT or AA vs PM2.5 mass concentration. Isn't the slope in the plots of Figure 4 equal to DTTm and AAm? This could be explicitly discussed and aid in the interpretation. Also, contrast this DTTm and AAm to the data in Table 1 which give a different trend in AAm (I believe), likely due to the intercept not affecting AAm from the slope method. This should be discussed.

Response 11: Figure 4 shows the OP obtained by the AA and DTT assays and normalised to sampled air volume ($OP_{AA,V}$ and $OP_{DTT,V}$) and not to the PM mass. The former option or atmospheric quantity is considered to have a closer relationship to human exposure.

Moreover, the correlation of PM-mass normalised OP data with PM mass would raise further questions. The slopes of the regression lines were explicitly given in lines 489–496 of the original MS. They were further discussed now.

Does it make sense that the "toxicity" based on AA (ie, the slope in Fig 4b) is the least for the city center and higher at the rural site? The term toxicity is used loosely here since these acellular assays really do not provide insight on actual toxicity. One might consider this when using the term toxicity throughout the manuscript. The AAm spatial trend is opposite other studies from Europe and most other regions; cities have higher mass normalized OP than rural areas. Are there studies that show the opposite, as found for AAm in this study? Do the authors think this is actually true from a health perspective; ie, the toxicity of rural aerosol, which is largely due to the influence of long-range transported dust, is more toxic than urban aerosols?

Response 12: The term toxicity was replaced by OP or larger sensitivity of OP determined by a particular assay in many relevant cases in the MS. The OP values normalised to sampled volume tend to increase from the rural area to the urban sites as indicated in Fig. 1, which is in line with other European studies. The situation could be complicated with the different contributions of BB and road vehicle sources at the different sites and seasons. See also response 11.

On both scatter plots, why not make the symbols, regression line and the r2 colors match? This would make it easier to interpret.

Response 13: The experimental data points have lighter shade of a colour, whereas the corresponding regression lines and coefficients of correlation have the darker shade of the same colour to separate the measured data (symbols) from the modelled quantities (lines and numbers).

Line 562, could one find a better word then pronouncedly?

Response 14: The word was deleted.

Finally, two different acellular assays were used. They respond differently to chemical species in the aerosol particles, and so respond differently to sources, which results in differing interpretations of the PM2.5 air quality in the various regions. It is concluded that more than one assay is needed to get a full picture of air quality. But more could be added. Are these two assays optimal for that goal, or can the authors suggest other pairs based on published findings. If not, what test should be done to select optimal assays? It seems like more interpretation could be gleaned from this study.

Response 15: More precisely, we suggested that multiple, at least two OP assays are to be deployed to get a more holistic picture (line 341 of the original MS). Other assay combinations are also possible; however, most researchers select the AA and DTT assays if only two methods are utilised. A reasonable compromise among the number of the assays,

time or labour demands of the experiments and expected results should be reached. Our next research project is to deal with these issues in a systematic manner, and the preliminary results indicate that further firm and conclusive interpretation are expected in this field. A brief extension was amended on this in section 3.6 Limitations and later possibilities.

Imre Salma
corresponding author